# Antimicrobial Activity of Various Disinfectants to Clean Thermoplastic Polymeric Appliances in Orthodontics

**DOI:** 10.3390/polym14112256

**Published:** 2022-05-31

**Authors:** Kanket Kiatwarawut, Dinesh Rokaya, Irin Sirisoontorn

**Affiliations:** Department of Clinical Dentistry, Walailak University International College of Dentistry (WUICD), Walailak University, 87 Ranong 2 Road, Dusit, Bangkok 10300, Thailand; kanket.kk@gmail.com (K.K.); dinesh.ro@wu.ac.th (D.R.)

**Keywords:** chemical disinfectant, clear aligner, clear retainer, orthodontics, systematic review, thermoplastic polymer, appliance

## Abstract

There is a lack of research evidence on the risk–benefits of the various disinfectants in cleaning products and cleansing regimens. This systematic review compared the antimicrobial activity of various chemical disinfectants to disinfect the thermoplastic polymeric appliances in orthodontics. The study was carried out using the Preferred Reporting Items for Systematic Reviews and Meta-Analyses (PRISMA) and Cochrane Handbook of Systematic Reviews of Interventions. An electronic search was conducted on Pubmed, Google Scholar, Scopus, ScienceDirect, and Springer. Two authors independently investigated the risk of bias in duplication. A total of 225 articles were collected. After removing duplicates, 221 articles remained, and after filtering their titles and abstracts, 11 articles met eligibility qualifications remained. Finally, nine articles that met the criteria were selected. It showed that both over-the-counter orthodontic appliance cleaners and applied-chemical disinfectants were effective against bacteria. The duration and frequency of usage guidelines cannot be concluded.

## 1. Introduction

The demands for esthetic treatment outcomes have recently increased [1], and the use of esthetic appliances during treatments has also extended. These have led the manufacturers to develop systems that are appealing to the patients, with an underlying goal of reducing appliance visibility [2]. Clear aligners gradually move teeth into an ideal position through computerized technology while minimizing microbial risk [3,4], dental trauma, and root resorption [5]. The orthodontic aligner protocol consists of 20–24 h of use, removal during meals, and brushing before re-wearing [4]. Similar instructions apply to thermoplastic clear retainers, a type of removable appliance that has grown in popularity due to its esthetic and translucency [6,7,8]. Wearing for a long time helps reduce relapse, while relapses can be influenced by a variety of factors [9,10].

Some studies on thermoplastic orthodontic appliances (TOA) revealed an increase in *S*. *mutans* and *Lactobacillus* spp. [11]. Alshatti [12] mentioned that the incidence and severity of white spot lesions were not significantly different among clear aligners, self-ligating brackets, and conventional brackets. On the other hand, it is reported that patients showed severe gingival inflammation and tooth decay after 4 months of eating and drinking without cleaning the appliance [13]. Thus, cleaning/disinfection of TOAs is important to maintain oral health and hygiene. For the chemical cleaning of prostheses or appliances, a variety of cleaning tablets are available, most of which are peroxide-generating in nature. These tablets are used in several studies [14,15,16] and are one of the most used remedies. Axe et al. [17] discovered various other household products used in different parts of the world for removable appliance cleaning/disinfection, with many such regimens recommended by dentists, prosthodontists, orthodontists, and other dental health care professionals. Over-the-counter mouthwashes, liquid hand soaps, vinegar, dishwashing detergents, salt, bicarbonate of soda, and plain water are examples of such products. Among various disinfectants, chlorhexidine has gained popularity [18] and is easy to use and has a pleasant smell. Corega^®^, Kukis^®^, Retainer Brite^®^, Invisalign Cleaning-Crystal Solution, etc. are examples and are available in the market. However, chlorhexidine can cause staining and unfavorable taste.

Chemical disinfectants can use corrosion of the thermoplastic appliances and change in color, resulting in shorter service life. It is difficult to determine the optimum disinfection time interval of chemical disinfectants [19], and most often, the treatments are not aligned with the consumer use patterns, making it difficult in regard to cleaning patterns [20,21]. Clear intraoral appliances are disinfected using a variety of cleaning procedures and chemicals, although the efficacy of these methods and chemicals remains debatable. Only a few studies systematically compared various cleaning treatments [22,23]. Charavet et al. [23] performed the study only on the cleaning and disinfection protocol of clear aligners, but the retainers were not included. In addition, there is a lack of systematic research evidence on the risk–benefit profile of the most commonly used cleaning products and regimens used in orthodontics. Hence, this is the first systematic evaluation of the effectiveness of antimicrobial agents that covers all clear transparent thermoplastic appliances such as invisible aligners and retainers. In addition, the review focuses on the effectiveness of these agents and methods of evaluation. This risk–benefit profile was also determined by assessing the antimicrobial effects of those cleaning products.

## 2. Materials and Methods

### 2.1. Protocol Registration

This systematic review was registered in the International Prospective Register of Systematic Reviews (PROSPERO) (CRD42018108564).

### 2.2. Search Strategy

This review was carried out using the Preferred Reporting Items for Systematic Reviews and Meta-Analyses (PRISMA) and the Cochrane Handbook of Systematic Reviews of Interventions [24]. An electronic literature search was performed in the Pubmed, Google Scholar, Scopus, ScienceDirect, and Springer from January 1966 to January 2022 in all fields using the following keywords: (disinfect) and (thermoplastic) and (orthodontic) and (appliance or retainer or aligner) and (plaque or biofilm). The keywords used for each database search are shown in Table 1.

Articles published in English were included in this study. Duplicates were removed from the original search results using the Endnote program (version X9, Clarivate Analytics, Philadelphia, PA, USA). The search protocol was performed using the PICO principle (Patient/Problem/Population; Intervention/Exposure; Comparison and Outcomes).

All types of invisible orthodontic appliances, such as copolyester, polypropylene, polyurethane, etc., were included. The appliances were then required to be treated with a chemical disinfectant to evaluate the cleaning process’s effectiveness. Both direct and indirect microorganism estimation could be considered.

### 2.3. Eligibility Criteria

The review question was formulated as “What is the most effective cleaning agent, and the method for determining its level of effectiveness?” Population, Intervention, Comparison, Outcome, and Study (PICOS) format was utilized in this study, as explained below.P: Population/Problem: Fabricated clear thermoplastic aligners or retainersI: Intervention: Cleaning and method of evaluationC: Controls: Negative controlO: Outcomes: Amount of bacterial reductionS: Study designs: RCTs, Clinical Control Trials (CCTs), retrospective controlled cohort studies, retrospective uncontrolled cohort studies, case reports, and laboratory studies.

Using the inclusion and exclusion criteria listed in Table 2 and Table 3, reviewers independently selected abstracts for the disinfection of thermoplastic orthodontic appliances. Initially, all the articles searched were vetted based on their titles, resulting in the exclusion of potentially irrelevant items. The selected studies were required to show the result of decolonization in terms of quantity or quality of microorganism reduction. The studies excluded were animal studies, finite element studies, descriptive studies, review articles, systematic reviews, and meta-analyses. The disagreement between the reviewers was settled through discussion. Then, the abstracts were read and examined by the qualifying criteria, and the final articles were then chosen. In the case of insufficient data regarding the articles, the reviewers contacted the authors.

### 2.4. Risk of Bias in Individual Studies

**Two review authors independently investigated the risk of bias in duplication**. Disagreements were resolved in this case by contacting the third review author. The recently revised Cochrane risk of bias method was used to assess the risk of bias in the included RCTs (R.O.B 2.0). The Cochrane Handbook suggests using the ROBINS-I tool to assess NRS [25]. Following that, we attempted to contact the authors in case additional information concerning their trials was required.

## 3. Results

### 3.1. Study Selection and Identification

After removing duplicates with EndNote, 221 results remained, which were then filtered based on their titles and abstracts. As a result, the remaining 11 articles were inspected on a full-text scale, and two articles were eliminated since they did not meet the eligibility qualification. Finally, nine articles were selected as they met the requirements (Figure 1).

### 3.2. Study Characteristics

Table 3 lists the features of the nine studies that were chosen, each of which was conducted between 2013 and 2019. Most of the studies (4 articles) were prospective studies. The included studies, primarily concerning clear aligners, demanded that the participants change their assigned chemical disinfectants and intervention every 2 weeks and collect specimens after the intervention. There was also one randomized controlled trial (RCT) with a split-mouth study that included Essix thermoplastic appliances, published in 2016 by Albana et al. [26], showing the effectiveness of orthodontic appliance cleaners.

### 3.3. Thermoplastic Orthodontic Material

The materials used in the literature significantly vary. Although four articles referred to the same manufacturer, Essix material [26,31,33,34], each of them inspected different types of Essix material, ranging from Essix type A, Essix type C+, and Essix ACE. Furthermore, only two studies focused on clear aligners, Invisalign [28,32], using polyurethane material, a distinguishing material apart from the previous material. There was no study on the Vivera^®^ retainer, which is made of polyurethane similar to the Invisalign clear aligner, despite different levels of thickness. Moreover, the type of invisible appliance studied by Levrini et al. [32] was unidentified.

### 3.4. Chemical Disinfectant

The chemical disinfectants used also vary. They, nonetheless, can be classified into two groups; The first, over-the-counter orthodontic appliance cleaners (OCC), and the second, applied-chemical orthodontic appliance cleaners (ACC). Examples of OCC include Fresh Guard^®^, Invisalign^®^ Cleaning Crystals, Corega^®^, Retainer Brite^®^, etc. Among these disinfectants, researchers found no application of denture cleaner in cleansing transparent orthodontic appliances. Regarding the second group, however, the data showed that CHX, a broad-spectrum antimicrobial, was generally used in most cases, apart from other household solutions such as vinegar, which was rarely used on some occasions.

### 3.5. Disinfection Protocol

It is interesting to see the different protocols applied in each study. For example, in Albanna et al. [26], the researcher began by brushing appliances with toothpaste before immersing them in water infused with a cleaning tablet. In contrast, in Akgün et al. [27], the researcher submerged the appliances in a chemical solution before brushing. Furthermore, the mechanical cleaning of retainers or aligners, either before or after immersing appliances in an antimicrobial solution, was not mentioned in certain studies. Another interesting aspect is that even the same type of effervescent tablet can be used for different time lengths. For instance, Corega^®^ can be used for either 5 or 15 min [26,27] with the same frequency of immersion, once daily, except in the CHX mouthwash group, where once every 4 days is recommended, according to Ismah et al. [34].

### 3.6. Microorganism Reduction Evaluation

Most of the studied microorganisms are total bacteria and *Streptococcus mutants*, the main bacteria leading to Caries disease. Two studies tried to identify different types of microorganisms [26,30]. An antimicrobial assessment can be carried out either with direct techniques such as a colony count [27,29,30,31,34] or an indirect technique, namely an optical density measurement using a microplate reader.

### 3.7. Effectiveness of Disinfectant

Most studies follow the same pattern, demonstrating that both OOC and ACC products can effectively eliminate accumulating bacteria/diseases on thermoplastic appliances. Nonetheless, the split-mouth RCT study found no statistical significance between chemical cleaning and mechanical cleaning.

## 4. Discussion

In this study, we evaluated the effectiveness of antimicrobial agents that covered all clear transparent thermoplastic appliances and evaluated the effectiveness of these agents and methods of evaluation. This risk–benefit profile was also determined by assessing the antimicrobial effects of those cleaning products. This research might serve as advice for dentists in recommending treatments to their patients that could help avoid dental caries and/or periodontal disease.

### 4.1. Transparent Orthodontic Appliance Material

The types of materials used in the nine selected articles differ, and this may affect the adherence quantity and accumulation of the intraoral microorganism. It may have an impact on the performance of both physical and chemical cleaning methods because certain materials may contain niches that benefit the hidden bacteria. Low et al. (2010) also discovered that fingerprint patterns of polyurethane, the main ingredient in Invisalign, benefit the initial biofilm formation, whether coccal or rod species [35]. Aside from that, the polycarbonate-based material was found to be stainless steel than the polyurenate-based material [36]. Furthermore, intraoral use may alter surface morphology and change chemical and mechanical properties [37] as a result of an increase in colonization rate.

### 4.2. Changes in Physical Properties

The physical properties of the materials used are critical for establishing a successful orthodontic treatment, both in terms of tooth movement and retention, because mechanical or chemical cleaning may cause scratching on a material surface. According to a study that used different types of chemical cleaners for 6 consecutive months, Retainer Brite^®^ could most effectively affect surface roughness when synthesizing an Essix C+ retainer made of polypropylene/ethylene. Furthermore, the presence of 3% hydrogen peroxide can alter flexural modulus [19]. Studies on polyurethane found that Invisalign^®^ cleaning crystal, Polident^®^, and Listerine^®^ can cause the most changes in light transmittance. However, there is no article concerning changes in physical properties according to this systematic review.

### 4.3. Chemical Disinfectant

Brushing is widely accepted as a method of cleaning removable appliances, according to the Dental Professional Recommendation, even though brushing with or without toothpaste can still increase surface roughness [38]. There is currently no gold standard for cleaning dentures or removable orthodontic appliances, and mechanical cleaning alone cannot completely remove cariogenic and periodontal pathogens. In addition, wearing a full cuspal coverage intraoral appliance for nearly 24 h a day can reduce salivary flow and enhance the protective cover for bacteria. As such, an included chemical should help decrease pathogens, despite Albanna et al. [26] reporting that mechanical brushing has no effect when compared to its chemical counterpart. However, in an ACC group, CHX mouthwash was shown to acquire a more unique ability than other disinfectants, as CHX is a cationic compound that has been shown to bind to salivary proteins through electrostatic interactions, and if the retainer is immersed in CHX mouthwash for a certain time, CHX [18] can disinfect as well as prevent bacterial colonization. Nevertheless, there has been no research on the maximum bactericidal concentration (MBC) of ACC group products to determine if they are suitable for (denture) cleaning.

For vinegar or acetic acid, when bacteria are exposed to low-acidity acids, they are more susceptible than they would otherwise be, and this has long been recognized. They are considered to have several mechanisms for causing toxicity. Because of the balance between their ionized and non-ionized forms, weak acids may permeate bacterial membranes more easily than strong acids. The non-ionized form can freely diffuse across hydrophobic membranes [39]. Consequently, liberated anions (in this case, acetate) tend to collapse the proton gradients required for ATP synthesis because they interact with the electron transport chain-pumped out periplasmic protons and shuttle them across the membrane again without passing through F1Fo ATP synthetase. Acid-induced protein unfolding and membrane and DNA damage may occur because the cell’s internal pH (usually around pH 7.6 [40,41] in neutralophilic bacteria) is greater than the external acid solution’s pH (normally around pH 5.8). As a distinct source of toxicity, the anion generated by this mechanism is the result of a range of events, including osmotic stress on the cell. As a result, different weak acids at the same pH can have a wide range of toxic effects on cells, depending on the anion’s nature, which is known but not fully understood [42,43,44].

Most OCC products contain a sulfate or carbonate group, which are alkalizing agents that aid in pH buffering. It can be hypothesized that variations in the effectiveness of appliance plaque removal by two chemical methods are due to their different mechanism of action. As an active gradient, sodium perborate is used in the cleaning tablet. Sodium perborate buffers H_2_O_2_ to a pH of about 10 in a saturated aqueous solution. Oxygen is liberated during the oxidation of H_2_O_2_. The effervescing action of the cleaner solutions is thought to be related to the evolved O_2_, which is supposed to have a mechanical cleaning effect [45]. Different materials are used to make various products. The citric acid in a cleaning tablet, for example, reacts with sodium bicarbonate to form washing soda, which is ideal for removing biofilm from material surfaces.

### 4.4. Microbial Reduction Evaluation

There are several options for measuring bacteria reduction based on the data collected. One of the most fundamental methods in bacteria count or colony count, which can only measure actual bacteria when a concentration of harvested bacteria is diluted to the point where the separation of colonies is visible and thus countable. Another indirect technique measurement, namely the optical density at 595 mm, is simple and quick, whereas a required step of staining with violet, or other alternatives, may cause dye stains on the extracellular matrix or thermoplastic material due to surface roughness. As a result, this method cannot be used to measure microorganisms directly. Another study using SEM to examine the decreasing density of bacteria can only present qualitative data, not quantitative data [29]. Furthermore, neither of the options can distinguish between dead and live pathogens. A study found that using the LIVE/DEAD BacLight Bacterial Viability Kit (Life Technologies, Switzerland) in conjunction with flow cytometry and a confocal microscope can show both dead and live bacteria as well as quantitative data [46]. Using two or more evaluation methods can lead to more accurate results. The downside of crystal violet staining is that both living and dead bacteria, including extracellular polymeric substances (EPS) [47,48], are slimes composed mainly of polysaccharides, proteins, and DNA and biosynthesized by several strains of microorganisms. LIVE/DEAD staining, a kind of fluorescence stain, can reduce the weakness of crystal violet, which can confirm the live and dead bacteria.

### 4.5. Limitation

Language bias could be one of the constraints, as evidenced by the exclusion of trials published in languages other than English. Access to specific databases was limited, and the authors may have missed several studies that could have aided this research. Furthermore, the articles under consideration lack homogeneity in terms of study design, materials and methods, and evaluation method. More RCTs will improve future systematic reviews, and a broader list of OCC and ACC samples/products will facilitate the researcher in evaluating the effectiveness and efficacy of chemical disinfectants.

## 5. Conclusions

Both the over-the-counter orthodontic appliance cleaners (OCC) and applied-chemical orthodontic appliance cleaners (ACC) have antimicrobial activity, but the effectiveness is still incomparable due to non-homogenous in terms of study design, materials and methods, and evaluation method. The question of “which disinfectant is the most efficient?” remains unanswered. The gold standard for cleaning thermoplastic orthodontic appliances has yet to be established.

## Figures and Tables

**Figure 1 polymers-14-02256-f001:**
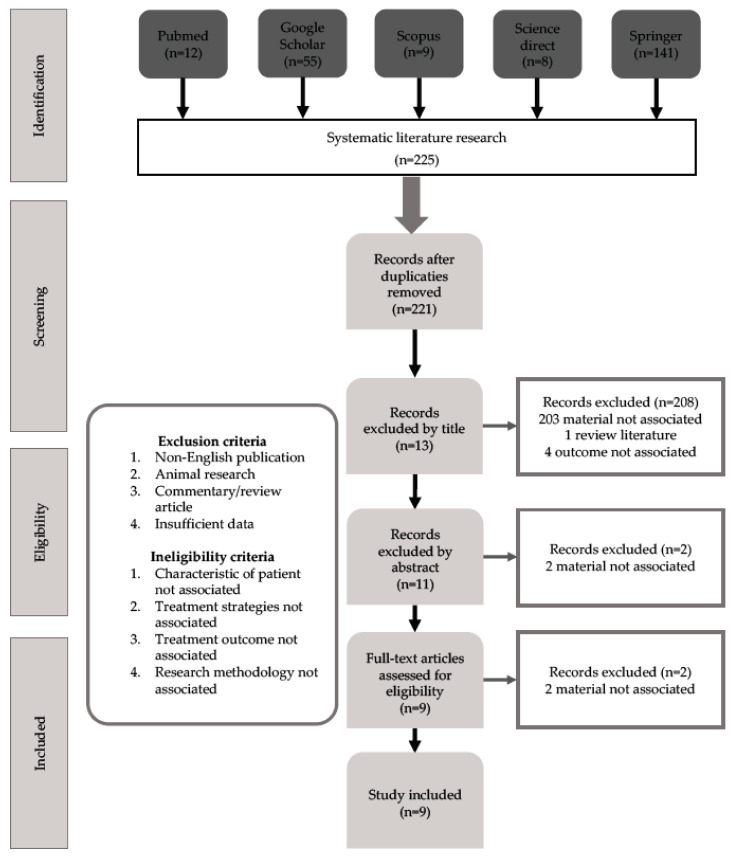
Prisma Flow Chart.

**Table 1 polymers-14-02256-t001:** Keywords used for each database search.

Database	Search Term
Pubmed	(‘‘Disinfection’’ OR ‘‘Cleaning’’ OR ‘‘Antimicroorganism’’ OR ‘‘Killing’’ OR “Decolonization”) AND (“Microbial” OR “Bacteria” OR ‘‘Microorganism’’ OR “Plaque” OR “Biofilm”) AND (“Vacuum forming” OR “Clear” OR “Invisible” OR “Thermoplastic” OR “Transparent”) AND (“Orthodontic”) AND (“Appliance” OR “Retainer” OR “Aligner”)
Google Scholar	(cleaning) and (retainer or aligner or clear orthodontic appliance)
Scopus	ALL (cleaning AND clear AND orthodontic AND appliance)
ScienceDirect	(cleaning) and (retainer or aligner or clear orthodontic appliance)
Springer	“cleaning” AND “clear” AND “orthodontic” AND “appliance”

**Table 2 polymers-14-02256-t002:** Selection criteria of the articles.

Inclusion Criteria	Exclusion Criteria	Ineligibility Criteria
Randomized controlled trials (RCTs) Prospective studies Retrospective studies Case series, case report Laboratory study	Non-English publication Animal research Commentary/review article Insufficient data (e.g., repeated abstract from the same paper, the thesis that found published paper, repeated samples by multiple publications, and statistical technique problems)	Characteristics of the patient not associated Treatment strategies not associated Treatment outcome not associated Research methodology not associated

**Table 3 polymers-14-02256-t003:** Overview of included studies.

Effectiveness of Disinfectant	Method of Microorganism Reduction Evaluation	Microorganism	Frequency	Disinfection Time	Cleaning Protocol	Chemical Disinfectant	Appliance Material	Type of Study	Author/Year
No significant differences between cleaning tablets and mechanical cleaning only	Bacterial Quantification with AlamarBlue^®^ Assay	Total bacteria	Once a day before bedtime	15 min	Step 1: one minute brushing with toothbrush and water before bedtime Step 2: Soaking in dissolved cleaning tablet in 150 mL tap water Step 3: Washing with tap water	1. Corega^®^ (GlaxoSmithKline, Dublin, Ireland) 2. Kukis^®^ (Procter & Gamble Technical Center Ltd., Egham, UK) 3. Retainer Brite^®^ (DENTSPLY, Bradenton, Fla)	Essix material (Invisacryl A, 0.030-inch, round, 0.75 mm/125 mm)	Randomized clinical trial (Split mouth study)	Albanna et al. (2016) [26]
No significant difference between cleaning tablets and vinegar but bacteria counts were statistically lower than in the control method	Colony count by colony-forming unit per 1 ml	*Streptococcus mutans* (SM) *Lactobacillus* (LB)	Once a day in the evening	5 min	Step 1: Keep the retainer in the cleaning solution Step 2: Brush with a soft brush and rinse with running water	1. Corega^®^ (GlaxoSmithKline, Brentford, Middlesex, United Kingdom) (First 2 weeks) 2. Water (control) (Next 2 weeks) 3. 5% white vinegar (Ferfresh, Fersan, Izmir, Turkey) (Last 2 weeks)	DispoDent Sert Gece Plagi, Yagmur Dental	Prospective study with a cross-over design	Akgün et al. (2019) [27]
Maximal reduction in biofilm accumulation was obtained when immersed in CS with a vibrating bath	Photodensitometer with 1 % gentian violet staining	Total bacteria	Once a day	15 min	Stage 1: Brush with 1400 ppm toothpaste (two aligners) Stage 2: Brush and immerse in CHX mouthwash and rinse water (70 days) Stage 3: Immerse in vibrating bath with CS and rinse with water	1. 0.12% chlorhexidine (CHX) mouthwash (Laser Co., Barcelona, Spain) 2. Invisalign Cleaning-Crystal solution (CS) (Align Technology, Santa Clara, CA, USA) Santa Clara, Calif	Thermoplastic material (Align Technology, Santa Clara, CA, USA)	Prospective study	Shpack et al. (2013) [28]
Effervescent tablets with brushing can reduce bacterial accumulation significantly	Scanning Electron microscope analysis	TotalScanningia	Once a day	Water group: 15 s Tablet group: 30 min Brushing group: 30 s	Stage 1: Rinse in running water for 15 s at least twice a day (two weeks) Stage 2: Soak in dissolved cleaning tablets, then brush with toothbrush and toothpaste at least 30 s Stage 3: Brush with toothpaste at least 30 s	1. Cold running water (control) 2. Effervescent tablets containing sodium carbonate and sulfate (Invisalign^®^ Cleaning System, Align Technology, San Joe, CA, USA) 3. Toothbrush and toothpaste, Fla)	Unknown	Prospective study	Levrini et al. (2015) [29]
No bacteria and fungal were found in the Cupral group	Colony count by colony-forming unit	Total bacteria and fungal load	1 time	1 h	Dissolve in saline buffer or 1.25% Cupral at room temperature	1. Saline buffer (control) 2. 1.25% Cupral (Humanchemie GmbH, Alfeld, Germany)	Novula (Rome, Italy)	Case report	Meto et al. (2019) [30]
All three cleaning methods removed 99% of microorganisms from the Essix orthodontic retainers	Colony count by colony forming unit	Methicillin-resistant *Staphylococcus aureus-16(MRSA-16)* *Streptococcus sanguinis* *Candida albicans* *Actinomyces naeslundii*	1 time	Group 1: 30 s Group 2: 30 s Group 3: 10 min Group 4: 10 min	Group 1: Brush with toothpaste Group 2: Brush with CHX Group 3: Immerse in CHX solution Group 4: Immerse in phosphate-buffered saline	1. Colgate cavity protection fluoride toothpaste (Group 1) 2. CHX gluconade gel (Corsodyl Dental Gel) (Group 2) 3. CHX solution (Corsodyl Alcohol-Free Mint Mouthwash) (Group 3) 4.Phosphate-buffered saline (Control group)	Essix ACE^TM^ (Dentsply Sirona, Charlotte, NC, USA)	In vitro laboratory study	Chang et al. (2013) [31]
The use of sodium carbonate and sulfate effervescent tablets combined with the mechanical debridement resulted in being the most effective method	Bioluminometer Microbiological Analysis	Total bacteria	Once a day	Group 1: 15 s Group 2: at least 30 s Group 3: 20 min	Group 1: Rinse in cold running water before eating Group 2: brush before eating Group 3: Soak in dissolved tablets and brush with toothpaste	1. Running water (Group 1) 2. Toothpaste (Group 2) 3. Effervescent tablets containing sodium carbonate and sulfate (Invisalign^®^ Cleaning System, Align Technology, San Joe, CA, USA) (Group 3)	Invisalign (Align Technology, Santa Clara, CA, USA)	Prospective study	Levrini et al. (2016) [32]
The effective cleaning tablets can be ordered as follow; Retainer Brite^®^, Smart guard^®^, Invisalign^®^ Cleaning Crystals, and Fresh Guard^®^	Optical density by spectrophotometer	Total bacteria	1 time	Group 1: 15 min Group 2: 5 min Group 3: 15–20 min Group 4: 20 min	Soak in cleaning tablet and follow the instruction	1. Invisalign^®^ Cleaning Crystals (Align Technology, Santa Clara, CA, USA) (Group 1) 2. Fresh Guard^®^ (Efferdent^®^) (Group 2) 3. Retainer Brite^®^ (Densply serona) (Group 3) 4.Smart guard^®^ Retainer and Aligner Cleaner (Smart guard^®^) (Group 4)	Essix A+^®^ Plastic (Dentsply Sirona Charlotte, NC, USA)	In vitro laboratory study	Pilloni (2019) [33]
Immersion in cleaning tablet once a day or in CHX solution twice per week will have the same efficacy for cleaning a thermoplastic retainer	Colony count by colony forming unit per millilitre	*Streptococcus mutans*	Group 1 and control group: once a day Group 2: once every 4 days	Group 1 and control group: 5 min Group 2: 10 min	Soak in cleaning solution	1. Denture disinfectant tablet solution (Polident) (Group 1) 2. 0.1% CHX mouthwash (Minosep) (Group 2) 3. Aqua Dest water (Control group)	Essix C+^®^ Plastic (Dentsply Sirona, Charlotte, NC, USA)	Clinical and laboratory experiment	Ismah et al. (2019) [34]

## Data Availability

Not applicable.

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
