# Peer review of "Antimicrobial Activity of Various Disinfectants to Clean Thermoplastic Polymeric Appliances in Orthodontics"

_polymers, 2022, doi:10.3390/polym14112256_

Round 1

Reviewer 1 Report

The manuscript by the Authors reports “Antimicrobial activity of various disinfectants to clean thermo-plastic polymeric appliances in orthodontics”. This clinical case is under the scope of this Materials Journal, the topic is relevant for readers and this research deals with potentially significant knowledge to the field.

However, there are some concerns about the present manuscript:

(Keywords)  

  • Please order the keywords alphabetically for a standardised presentation of the keywords.

(Introduction)

Statement of Clinical Relevance

  • What is the importance of this review for the clinical? 
  • What this study has new?
  • (Rationale) Describe the rationale for the review in the context of what is already known.
  • (Objectives), At this point, the authors must provide an explicit statement of questions being addressed with reference to participants, interventions, comparisons, outcomes, and study design (PICOS).

(Discussion)

  • Please, identified what was the strength(s) of this study? And also, implications for future perspectives.
  • Evidence assessment of best practices: What do the Authors recommend? What is the flowchart action for this situation?

References

  • But references are not standardized. The titles of references have a different format, the title of the article is written in capital letters at the beginning of words, others only in lower case. Also, the standardized format of presentation in the journal's name. Because names have been written in a different formats, one is not abbreviated, and others are not.

Author Response

Response to Reviewer 1 Comments

The manuscript by the Authors reports “Antimicrobial activity of various disinfectants to clean thermos-plastic polymeric applications in Orthodontics”. This clinical case is under the scope of this Materials Journal, the topic is relevant for readers and this research deals with potentially significant knowledge to the field.

However, there are some concerns about the present manuscript:

Thank you for your positive comments. Corrections in the Manuscript for Reviewer 1 are highlighted in Yellow color.

(Keywords)  

  • Please order the keywords alphabetically for a standardised presentation of the keywords.

Response: The content of Keywords, is ordered alphabetically.

Chemical disinfectant; clear aligner; clear retainer; orthodontics; systematic review; thermoplastic polymer, appliance

(Introduction)

Statement of Clinical Relevance

  • What is the importance of this review for the clinical? 
  • What this study has new?
  • (Rationale) Describe the rationale for the review in the context of what is already known.
  • (Objectives), At this point, the authors must provide an explicit statement of questions being addressed with reference to participants, interventions, comparisons, outcomes, and study design (PICOS).

Response: The Introduction and Methods, is improved.

Clear intraoral equipment is disinfected using a variety of cleaning procedures and chemicals, although the efficacy of these methods and chemicals remains debatable. At the very least, this research might serve as a guideline for dentists in recommending treatments to their patients that could help avoid dental caries and/or periodontal disease.

  • This is the first systematic evaluation to document the effectiveness of antimicrobial agents to clean transparent thermoplastic equipment such as invisible aligners and retainers. The review focuses on the effectiveness of these agents and methods of evaluation
  • In comparison to other disinfectants, chlorhexidine has been found to gain a more specialized ability[1] but this agent is able to be a cause of staining. The taste is quite unfavorable. In addition, there are a variety of manufacturers. products that are competitive, easy to use and pleasant smelling; for example, Corega®, Kukis®, Retainer Brite®, Invisalign Cleaning-Crystal Solution etc. are all examples of this. Comparing the performance of products available on the market is therefore critical.
  • There is also research of chemical corrosion that can cause thermoplastic appliances to change color and have a shorter service life. For each chemical, it is critical to determine the optimum disinfection time intervals [2].
  • Participant =        Fabricated clear thermoplastic aligners or retainers
    Intervention   =        Immerse in disinfectant
    Comparison  =        Various chemical disinfectants
    Outcome       =        Amount bacterial reduction

(Discussion)

  • Please, identified what was the strength(s) of this study? And also, implications for future perspectives.
  • Evidence assessment of best practices: What do the Authors recommend? What is the flowchart action for this situation?

Response: The Discussion is improved.

Not only was the efficacy of cleaning products investigated in this study. There is also an assessment approach for bacterial decrease that is described here. Conventional methods such colony forming unit counts, which revealed a significant reduction in bacteria, were used to evaluate bacterial reduction. However, this is a time-consuming approach that is prone to human mistake [3]. Crystal violet staining is a simple but possibly false positive procedure because crystal violet is stained with an extracellular polymeric material (EPS) [4]. Fluorescence staining of bacteria, such as the LIVE/DEAD BacLight Bacterial Viability Kit (Life Technologies, Switzerland), is also being researched, which is a straightforward procedure but the type of dye must be particular to reduce the occurrence of false negative or false positive results[5]. Using two or more evaluation methods can lead to more accurate results. The downside of crystal violet staining is that both living and dead bacteria, including EPS, can be stained. LIVE/DEAD staining, a kind of fluorescence stain, can reduce the weakness of crystal violet which is able to confirm the live and dead bacteria.

References

  • But references are not standardized. The titles of references have a different format, the title of the article is written in capital letters at the beginning of words, others only in lower case. Also, the standardized format of presentation in the journal's name. Because names have been written in a different formats, one is not abbreviated, and others are not.

Response: The References are corrected.

Reviewer 2 Report

Comments 0512

  1. Abstract & conclusion: “both over-the-counter orthodontic appliance cleaners and 21 applied-chemical orthodontic appliance cleaners' effectiveness in responding to bacteria is deemed greater than or equal to mechanical cleaning.” What is conclusion based on?
  2. Material and method: There were two “in vitro laboratory studies” included in this paper, but two animal studies excluded. Please to explain the rationale.
  3. Results: please to provide more detailed information of these included papers.
  4. Discussion: please to compare antimicrobial activity of different cleaning agents much more than the present discussion portion.
  5. Conclusion: The question of "which disinfectant is the most efficient?" remains unanswered. Therefore, please to provide the clinical significance of this review.

Author Response

Response to Reviewer 2 Comments

Comments

Thank you for your positive comments. Corrections in the Manuscript for Reviewer 2 are highlighted in Green color.

  1. Abstract & conclusion: “both over-the-counter orthodontic appliance cleaners and 21 applied-chemical orthodontic appliance cleaners' effectiveness in responding to bacteria is deemed greater than or equal to mechanical cleaning.” What is the conclusion based on?

Response: The content of Abstract is improved

  1. Material and method: There were two “in vitro laboratory studies” included in this paper, but two animal studies excluded. Please to explain the rationale.

We exclude animal study because there are many factors that are different from human oral cavity such as pH, temperature, organic and inorganic constituents, as well as microflora[1]. These variables could be controlled in a laboratory study to the greatest extent possible to mimic the condition of the human oral cavity.

  1. Results: please to provide more detailed information of these included papers.

The more information about included articles is prepared in Table 3

  1. Discussion: please to compare antimicrobial activity of different cleaning agents much more than the present discussion portion.

Response: The content of Discussion is improved

CHX mouthwash was shown to acquire a more unique ability than other disinfectants, as CHX is a cationic compound that has been shown to bind to salivary proteins through electrostatic interactions, and if the retainer is immersed in CHX mouthwash for a certain time, CHX  can disinfect as well as prevent bacterial colonization. Nevertheless, there has been no research on the maximum bactericidal concentration (MBC) of ACC group products to determine if they are suitable for (denture) cleaning

It can be hypothesized that variations in the effectiveness for appliance plaque removal by two chemical methods due to their different mechanism of action. As an active gradient, sodium perborate is used in the cleaning tablet. Sodium perborate buffers H2O2 to a pH of about 10 in a saturated aqueous solution. Oxygen is liberated during the oxidation of H2O2. The effervescing action of the cleaner solutions is thought to be related to the evolved O2, which is supposed to have a mechanical cleaning effect[2]. In contrast, When bacteria are exposed to low-acidity acids, they are more susceptible than they would otherwise be, and this has long been recognized. They are considered to have a number of mechanisms for causing toxicity. Because of the balance between their ionized and non-ionized forms, weak acids may permeate bacterial membranes more easily than strong acids. The non-ionized form can freely diffuse across hydrophobic membranes[3]. Consequently, liberated anions (in this case acetate) tend to collapse the proton gradients required for ATP synthesis because they interact with the electron transport chain-pumped out periplasmic protons and shuttle them across the membrane again without passing through F1Fo ATP synthetse. Acid-induced protein unfolding and membrane and DNA damage may occur because the cell's internal pH (usually around pH 7.6[4,5] in neutralophilic bacteria) is greater than the external acid solution's pH (normally around pH 5.8) As a distinct source of toxicity, the anion generated by this mechanism is the result of a range of events including osmotic stress on the cell. As a result, different weak acids at the same pH can have a wide range of toxic effects on cells, depending on the anion's nature, which is known but not fully understood.[6-8]. Both our findings and those of Bjarnsholt et al., published only recently, are in line with and even anticipated by those of the aforementioned studies.

  1. Conclusion: The question of "which disinfectant is the most efficient?" remains unanswered. Therefore, please to provide the clinical significance of this review.

Response: The content of Conclusion is corrected

A clinical research comparing three hygenic protocol found that the best way to reduce microorganism is to use a vibrating bath with cleaning solution[9]. However, another RCT did not cliam that cleaning tablet or cleaning solution did not significantly reduce the bacterial count in retainers when compared to used of mechanical cleaning alone[10].

  1. Marsh, P.D. Role of the Oral Microflora in Health. Microbial Ecology in Health and Disease 2000, 12, 130-137, doi:10.1080/089106000750051800.
  2. Mueller, H.J.; Greener, E.H. Characterization of some denture cleansers. The Journal of prosthetic dentistry 1980, 43, 491-496.
  3. Walter, A.; Gutknecht, J. Monocarboxylic acid permeation through lipid bilayer membranes. The Journal of membrane biology 1984, 77, 255-264.
  4. Slonczewski, J.L.; Rosen, B.P.; Alger, J.R.; Macnab, R.M. pH homeostasis in Escherichia coli: measurement by 31P nuclear magnetic resonance of methylphosphonate and phosphate. Proceedings of the National Academy of Sciences 1981, 78, 6271-6275.
  5. Slonczewski, J.L.; Fujisawa, M.; Dopson, M.; Krulwich, T.A. Cytoplasmic pH measurement and homeostasis in bacteria and archaea. Advances in microbial physiology 2009, 55, 1-317.
  6. Roe, A.J.; McLaggan, D.; Davidson, I.; O’Byrne, C.; Booth, I.R. Perturbation of anion balance during inhibition of growth of Escherichia coli by weak acids. J Bacteriol 1998, 180, 767-772.
  7. Hirshfield, I.N.; Terzulli, S.; O'Byrne, C. Weak organic acids: a panoply of effects on bacteria. Science progress 2003, 86, 245-270.
  8. Salmond, C.V.; Kroll, R.G.; Booth, I.R. The effect of food preservatives on pH homeostasis in Escherichia coli. Microbiology 1984, 130, 2845-2850.
  9. Shpack, N.; Greenstein, R.B.-N.; Gazit, D.; Sarig, R.; Vardimon, A.D. Efficacy of three hygienic protocols in reducing biofilm adherence to removable thermoplastic appliance. The Angle Orthodontist 2014, 84, 161-170.
  10. Albanna, R.H.; Farawanah, H.M.; Aldrees, A.M. Microbial evaluation of the effectiveness of different methods for cleansing clear orthodontic retainers: A randomized clinical trial. The Angle Orthodontist 2017, 87, 460-465.

Round 2

Reviewer 1 Report

This research is under the scope of this journal; the topic is relevant for readers, and this research deals with potentially significant knowledge of the field. 

The authors improved the quality of the manuscript after the reviewer's indications.

Author Response

Response to Reviewer 1 Comments

This research is under the scope of this journal; the topic is relevant for readers, and this research deals with potentially significant knowledge of the field. 

The authors improved the quality of the manuscript after the reviewer's indications.

Thank you for your positive response.

Reviewer 2 Report

1.      Abstract: The present systematic review concluded that both over-the-counter orthodontic appliance cleaners and applied-chemical orthodontic appliance cleaners' effectiveness in responding to bacteria is deemed unsummarizable. However, the duration and frequency of usage guidelines cannot be summarized.

è Please to confirm again.

2.      Introduction: In addition, only a few studies systematically compared various cleaning treatments and This is the first systematic evaluation to document the effectiveness of antimicrobial agents to clean transparent thermoplastic equipment such as invisible aligners and retainers. The review focuses on the effectiveness of these agents and methods of evaluation

è Please to re-check.

3.      Discussion 4.4.: Using two or more evaluation methods can lead to more accurate results. The downside of crystal violet staining is that both living and dead bacteria, including EPS, can be stained. LIVE/DEAD staining, a kind of fluorescence stain, can reduce the weakness of crystal violet which can confirm the live and dead bacteria.

è Please to add reference(s) and to explain EPS.

4.      Conclusion: Both OCC and ACC have antimicrobial activity but the effectiveness is still incomparable since, In terms of study design, materials and methods, and evaluation method, the publications under consideration are not homogenous. More RCTs will improve future systematic reviews. However, clinical research comparing three hygenic protocols found that the best way to reduce microorganisms is to use a vibrating bath with a cleaning solution

è Please to explain clearly.

Author Response

Thank you for your positive comments. Throughout corrections and English editing is done in the manuscript. The major corrections for Reviewer 2 are highlighted in Green color.

  1. Abstract: The present systematic review concluded that both over-the-counter orthodontic appliance cleaners and applied-chemical orthodontic appliance cleaners' effectiveness in responding to bacteria is deemed unsummarizable. However, the duration and frequency of usage guidelines cannot be summarized.

è Please to confirm again.

Response: The Abstract is edited and improved.

  1. Introduction: In addition, only a few studies systematically compared various cleaning treatments and This is the first systematic evaluation to document the effectiveness of antimicrobial agents to clean transparent thermoplastic equipment such as invisible aligners and retainers. The review focuses on the effectiveness of these agents and methods of evaluation

è Please to re-check.

Response: The contents in the Introduction are edited and improved

  1. Discussion 4.4.: Using two or more evaluation methods can lead to more accurate results. The downside of crystal violet staining is that both living and dead bacteria, including EPS, can be stained. LIVE/DEAD staining, a kind of fluorescence stain, can reduce the weakness of crystal violet which can confirm the live and dead bacteria.

è Please to add reference(s) and to explain EPS.

Response: The content in the Discussion is edited and improved

  1. Conclusion: Both OCC and ACC have antimicrobial activity but the effectiveness is still incomparable since, In terms of study design, materials and methods, and evaluation method, the publications under consideration are not homogenous. More RCTs will improve future systematic reviews. However, clinical research comparing three hygenic protocols found that the best way to reduce microorganisms is to use a vibrating bath with a cleaning solution

è Please to explain clearly.

Response: The Conclusion is edited and improved.

Round 3

Reviewer 2 Report

none